biomedical engineering/biomechanics

valve-in-valve (ViV), transcatheter aortic valve implantation (TAVI), computational fluid dynamics, leaflet thrombosis

**Author for correspondence:**
Romina Plitman Mayo
e-mail: romina.p@weizmann.ac.il

# Numerical models for assessing the risk of leaflet thrombosis post-transcatheter aortic valve-in-valve implantation

Romina Plitman Mayo[1,3], Halit Yaakobovich[2],
Ariel Finkelstein[4], Shawn C. Shadden[5] and Gil Marom[1]

[1]School of Mechanical Engineering and [2]Department of Biomedical Engineering,
Tel Aviv University, Tel Aviv, Israel
[3]Department of Biological Regulation, Weizmann Institute of Science, Herzl Street 234, Rehovot, Israel
[4]Division of Cardiology, Tel Aviv Medical Center, Tel Aviv, Israel
[5]Department of Mechanical Engineering, University of California, Berkeley, CA, USA

RP, 0000-0002-3013-0717; GM, 0000-0003-3130-3402

Leaflet thrombosis has been suggested as the reason for the reduced leaflet motion in cases of hypoattenuated leaflet thickening of bioprosthetic aortic valves. This work aimed to estimate the risk of leaflet thrombosis in two post-valve-in-valve (ViV) configurations, using five different numerical approaches. Realistic ViV configurations were calculated by modelling the deployments of the latest version of transcatheter aortic valve devices (Medtronic Evolut PRO, Edwards SAPIEN 3) in the surgical Sorin Mitroflow. Computational fluid dynamics simulations of blood flow followed the dry models. Lagrangian and Eulerian measures of near-wall stagnation were implemented by particle and concentration tracking, respectively, to estimate the thrombogenicity and to predict the risk locations. Most of the numerical approaches indicate a higher leaflet thrombosis risk in the Edwards SAPIEN 3 device because of its intra-annular implantation. The Eulerian approaches estimated high-risk locations in agreement with the wall sheer stress (WSS) separation points. On the other hand, the Lagrangian approaches predicted high-risk locations at the proximal regions of the leaflets matching the low WSS magnitude regions of both transcatheter aortic valve implantation models and reported clinical and experimental data. The proposed methods can help optimizing future designs of transcatheter aortic valves with minimal thrombotic risks.

# 1. Introduction

Transcatheter aortic valve implantation (TAVI) is a minimally invasive intervention whereby a bioprosthetic valve mounted on a stent is delivered and deployed on a stenotic valve. Although TAVI has recently been approved for low-surgical risk patients [1], its wide use is still constrained by the increasing evidence of subclinical leaflet thrombosis post-TAVI [2–4]. Subclinical leaflet thrombosis has been suggested as the underlying reason for hypoattenuated leaflet thickening (HALT) [5], leading to a reduced leaflet motion [2]. In fact, reduced leaflet motion was resolved in up to 100% of the patients receiving anticoagulants compared to only 9% of those who did not [2]. The highest occurrence of HALT is reported for patients with a valve-in-valve (ViV) implantation, whereby a TAVI is performed inside a degenerated surgical valve [3,4]. However, the exact prevalence of leaflet thrombosis remains unclear because it is clinically under-diagnosed [6].

Thrombosis is a common complication of cardiovascular diseases [7]. There are three main factors contributing to thrombus formation: hypercoagulability, endothelial dysfunction and haemodynamic changes [8]. Within the haemodynamic category, exposure of platelets to high flow stresses and platelet adhesion in stagnant flow regions are considered the two main mechanical causes [9,10].

Many causes have been suggested as reasons for thrombus formation after TAVI, including haemodynamic factors [11]. Low flow rate and regions of flow stagnation near the valve have been suggested as the source of leaflet thrombosis since they have been observed in many *in vitro* and *in silico* studies [12–17]. These studies demonstrated that the valve confinement by the previous leaflets increases blood stagnation within the neo-sinuses and near the leaflets. However, experiments usually simplify the aortic root (AR) and valve geometry so much that they bear little physiological significance. Additionally, capturing the flow patterns in the neo-sinuses has proved to be experimentally challenging. On the other hand, computational modelling can provide insight into the haemodynamic characteristics of the aortic valve with different procedural options.

Computational fluid dynamics (CFD) also has the capability to calculate and analyse the thrombosis-related flow characteristics in the entire flow field. Several different *in silico* approaches have been suggested to predict the risk and location of thrombus formation resulting from flow stagnation. Both *in silico* wall shear stress (WSS) and residence time (RT) measures have been correlated with the *in vivo* thrombus locations [18,19]. These measures can be calculated either by Lagrangian or Eulerian approaches [10,20,21]. The simplest approach is to analyse the WSS features from the CFD results [22]; specifically, by evaluating the WSS on the desired surface and locating the points where the WSS vector vanishes with an unidentified direction (WSS fixed points; WSSfp). These points have been regarded as accumulation points and have been correlated with high-concentration areas [23,24]. Additionally, the WSS streamlines have been used as signatures for near-wall transport in blood flow and have been validated using wall-particle image velocity experiments [25].

In Eulerian methods, the RT is calculated by explicitly solving partial differential equations [18]. Commonly, the advection-diffusion equation describes the accumulation of species in the blood domain as a continuum concentration fraction. A different approach is to use the advection-diffusion equation to characterize residence time [21], a method that has been previously used to model thrombus formation [26], flow stagnation in pulsatile ventricular assist devices and the left ventricle [20,27], and recently in aneurysms [21].

In Lagrangian approaches, particles are seeded and tracked while the flow characteristics are derived along their trajectories [28]. Two common Lagrangian estimates are the particle residence time [29,30] and the mean exposure time [21,31], whereby the time each particle spends inside the domain or the normalized cumulative time of all particles residing in a subdomain is calculated, respectively. The particles can be assumed not to affect the Eulerian flow field (continuum) or their trajectories can be calculated by coupling their solution with the Eulerian blood flow solution [10,21].

Because there are several thrombosis risk estimation approaches, each with different modelling assumptions, special care should be taken. The aim of this work is to employ and compare five different *in silico* approaches for the estimation of leaflet thrombosis and predict their high-risk locations. To this end, two models of ViV procedures were generated [32] and different numerical methods were employed to estimate leaflet thrombogenicity.

# 2. Methods

## 2.1. Computational models

The blood flow models post-ViV are based on geometries from our previous finite-element analysis studies of the ViV implantation [32]. Briefly, implantations of two TAVI devices—Medtronic Evolut

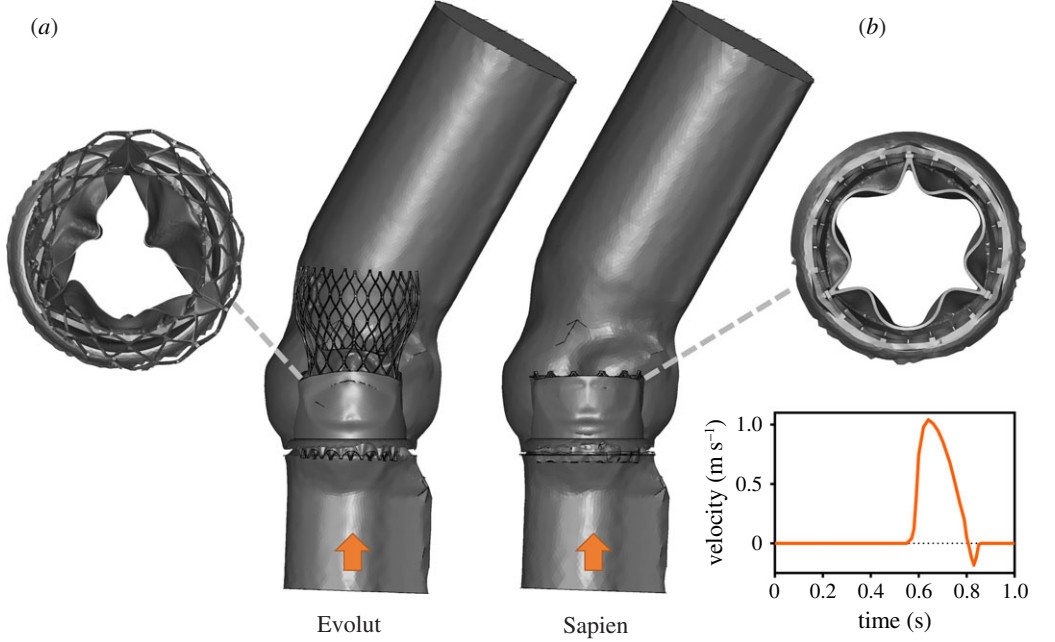

**Figure 1.** Fluid domains for the Evolut (*a*) and SAPIEN (*b*) ViV models. A top view of the TAVI leaflets systolic configuration is given, together with the velocity waveform prescribed at the left ventricular outflow tracts.

PRO and Edwards SAPIEN 3 (26 mm, [33])—inside a 27 mm Sorin Mitroflow surgical valve were simulated. These three-dimensional models were solved using ABAQUS EXPLICIT (Simulia, Dassault Systèmes) while considering only the stents of the TAVI devices. To calculate the position of the cuff and leaflets of each TAVI device, nodal displacement boundary conditions were applied to the stents of the deployed configurations with a linear elastic behaviour [34,35]. The leaflets position after the SAPIEN ViV implantation resembles its zero stressed configuration. The Evolut leaflets, which in their zero-stress configuration are closed, were opened into their systolic position by applying a pressure boundary condition of up to 0.26 mmHg. Note that the deformed Evolut stent is greatly influenced by the shape of the ascending aorta, leading to an asymmetric deployment of the stent and asymmetric configuration of the leaflets. The leaflets' mesh included 65 000 hexahedral elements and 71 000 three-dimensional elements for the SAPIEN and Evolut PRO models, respectively.

These systolic configurations represent the thrombogenicity worst-case scenario because the neo-sinuses are narrowest and deepest. CFD models were then generated and meshed using ANSYS Fluent Meshing (Release 2019 R3, ANSYS, Inc.). The fluid domain was derived from the post-procedural ViV geometries, as shown in figure 1. The fluid domains were meshed with approximately 7.1 million and 2.5 million tetrahedral cells for the Evolut and the SAPIEN models, respectively [34].

For each model, transient flow analysis was performed using ANSYS Fluent. The flow was assumed to be laminar and the blood to be Newtonian and incompressible with a density of $1060 \, \mathrm{kg^3 \, m^{-1}}$ and viscosity of $0.0035 \, \mathrm{kg \, m \cdot s^{-1}}$ [36]. A time-dependent velocity waveform [37] was prescribed at the inlet of the AR (figure 1) and a zero pressure condition was prescribed at the model outlet. A no-slip wall condition was set for all the other surfaces, unless specified otherwise. Because the model focused on the systole, the coronary flow was neglected. The models were solved for 10 cardiac cycles to ensure periodic solution with a time-step size of 1 ms.

## 2.2. Thrombogenic risk estimation

Five different numerical approaches were employed to assess the thrombogenic potential and high-risk locations owing to stagnation.

(i) *Wall shear stress fixed points (WSSfp):* the WSS vector field, defined as the tangential component of the traction vector on the surface, was directly obtained from the CFD simulations. The last cycle time-averaged WSS streamlines were plotted as line integral convolution on the leaflets' surface using PARAVIEW [38]. The stagnation points were determined following the classification proposed by Arzani *et al.* [23]. In our case, the points of interest are those where the WSS vector disappears

| SC | PA | VLI |
|---|---|---|
| flux from the leaflet surface | equally spaced injections - 10 µm from the leaflets | equally spaced injections - 3 hights in the neo-sinuses |

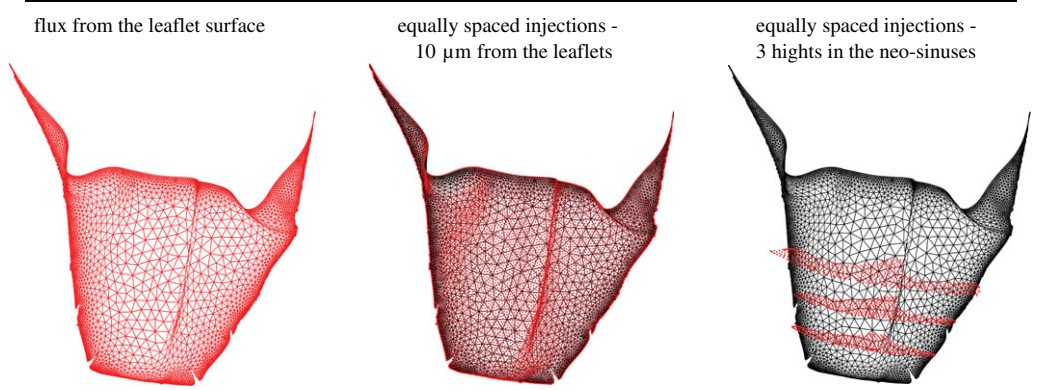

**Figure 2.** Boundary conditions of the different thrombogenicity approaches shown in red in the right coronary leaflet of the Evolut model: species flux from the leaflet surface (Species concentration, SC), an equally spaced seed of particles from a 10 µm distance from the leaflet (particle accumulation, PA) and an equally space seed of particles from three heights within the neo-sinuses (vortex location identification, VLI).

towards the surface (saddle points), since they have been shown to correlate with high-concentration zones and stagnation [23].

**Eulerian advection-diffusion approaches:** the accumulation of platelets or time was computed by solving the advection-diffusion equation coupled with the fluid momentum and mass conservation equations.

(ii) *Species concentration (SC):* the equation which describes the concentration of platelets in the blood flow is

$$\frac{\partial c}{\partial t} + \boldsymbol{u} \cdot \nabla c = \nabla \cdot (D \nabla c), \tag{2.1}$$

where $c$ is the platelet concentration. The platelets had the same material properties as the blood and a mass diffusivity ($D$) of $5 \times 10^{-10}$ m² s⁻¹. While the exact value of $D$ does not influence the results, it was chosen to ensure an advection dominating transport [21]. A constant flux of platelets was released from the TAVI leaflets (figure 2) during the second cardiac cycle and the resultant mesh nodes SC were saved every time-step. The SC results were summed up during the last cardiac cycle using an in-house post-processing MATLAB (MathWorks Inc., Natick, MA, USA) code to obtain the time-accumulated SC in a leaflet boundary layer of 0.25 mm [39]. These results were plotted on the leaflets' surface to identify the thrombus high-risk locations.

(iii) *Eulerian residence time (ERT):* the advection-diffusion equation which describes the ERT, represented by $c$, with a source term equal to one is

$$\frac{\partial c}{\partial t} + \boldsymbol{u} \cdot \nabla c = \nabla \cdot (D \nabla c) + 1. \tag{2.2}$$

Therefore, the source term represents the time that is being integrated in an Eulerian framework [21]. The last cycle ERT accumulation was calculated similar to the SC model.

**Lagrangian particle tracking approaches:** A Lagrangian–Eulerian approach, whereby platelet particles are seeded to the fluid and their trajectories are calculated within the CFD solver, was implemented. The trajectories were calculated by integrating the particle force balance equation within the Eulerian solver. The fluid phase influenced the particles *via* inertia and drag forces, while the particles influenced the fluid phase *via* source terms of mass, momentum, and energy [10]. The platelet particles were modelled as neutrally buoyant spherical particles with a diameter of 3 µm. Thrombogenicity was estimated by the two following approaches.

(iv) *Particle accumulation (PA):* approximately 35 000 and 26 000 particles were seeded near to the aortic side of the SAPIEN and Evolut leaflets, respectively. To avoid any bias owing to mesh heterogeneity, the particles were seeded from equally spaced locations rather than from the mesh nodes (figure 2). The particles were released at a distance of 10 µm from the TAVI leaflets with a slow initial velocity ($10^{-12}$ m s⁻¹), allowing them to be driven by the flow field. While the particles

were released at the beginning of the second cycle, their accumulation was calculated during the last cardiac cycle, accounting only for the particles that were not washed out. An in-house MATLAB post-processing code was employed to estimate the particle accumulation within a boundary layer of 0.25 mm [39]. To this end, the particles' shortest distance to the leaflets surface was calculated in every time-step and the closest surface element (facet) identified. The accumulation of particles was then obtained by adding all the particles residing within the boundary layer *per* element, over time.

(v) *Vortex location identification (VLI):* approximately 29 000 and 28 000 particles were released at three different heights (figure 2) from the neo-sinuses cavities. An in-house MATLAB code was used to identify the vortices inside the neo-sinuses from the particles' trajectories. Here, a vortex was defined by a particle that changes its axial direction at least six times, assuming that it represents three loops along their trajectory. Mathematically, a direction change is identified as a sign change in the axial (global forward direction) coordinate difference between the locations in sequential time-steps. Additionally, the VLI magnitude was defined as the ratio between the average vortex diameter and its centre distance to the TAVI leaflets. This ratio implies that closer and bigger vortices have a larger influence on the risk of leaflet thrombosis. To visualize the results, the closest surface element was identified and all the VLI values attributed to the same element were summed up.

# 3. Results

## 3.1. Comparison between the Evolut and SAPIEN models

Initially, the deployed configurations of the two TAVI devices were compared. Thereafter, the different thrombosis risk estimations were analysed and compared. In all the models, the results were calculated using the tenth cycle data and plotted in a spread view of the TAVI device leaflets [32]. The colour contours indicate the thrombogenic risk, with red being the higher risk locations and blue the lower risk ones. The black regions are those locations where the TAVI leaflets are in contact with its cuff and stent; therefore, these regions have no risk of thrombus formation.

### 3.1.1. Deployed configurations

The presentation of the TAVI leaflets in figure 3 demonstrates how the ViV implantation position leads to different confinements. This figure provides a spread view of the AR where the confined regions of the TAVI leaflets are shown in red. It is clear that the SAPIEN valve, which is usually implanted in an intra-annular position, is completely confined within the surgical valve. By contrast, the Evolut is only partially confined (52%) owing to its supra-annular implantation. This confinement colouring marks where the neo-sinuses are located and where the stagnation, and potential thrombus formation, is expected to occur.

### 3.1.2. Wall shear stress

The WSS magnitude and critical points, derived from the vector field, were initially compared. Figure 4 shows the line integral convolution (representing streamlines) of the WSS vectors on the surface of the TAVI leaflets, coloured by contours of the WSS magnitude. Low thrombogenic risk is associated with high WSS and is marked in blue. On the other hand, high thrombogenic risk is related to low WSS and is shown in red. Based on the WSS magnitude, it is noticeable that the deeper region of the pockets of the neo-sinuses has a higher risk of thrombosis. The SAPIEN leaflets are at a higher risk than the Evolut TAVI, especially the non-coronary and the left coronary leaflets. The Evolut leaflets experienced lower WSS magnitudes with a minimum of $1.44 \times 10^{-6}$ Pa versus $1.8 \times 10^{-6}$ Pa in the SAPIEN case. Magenta circles mark the WSSfp and, surprisingly, there are more of them in each Evolut leaflet than in the corresponding leaflet of the SAPIEN.

### 3.1.3. Eulerian approaches

The results of the two Eulerian approaches, shown in figure 5, indicate similar risky areas. It is worth noting that these contour plots are shown in their optimal colour scale to allow visualization of the data spatial distribution; the dashed lines show the leaflets' free-edges to indicate the confinement by the surgical valve. Both approaches predicted the risky areas to be on both sides of the central fold of the SAPIEN leaflets. The high-risk locations predicted by the SC and ERT approaches seem to better match the WSSfp than the low WSS magnitudes shown in figure 4. It should be mentioned

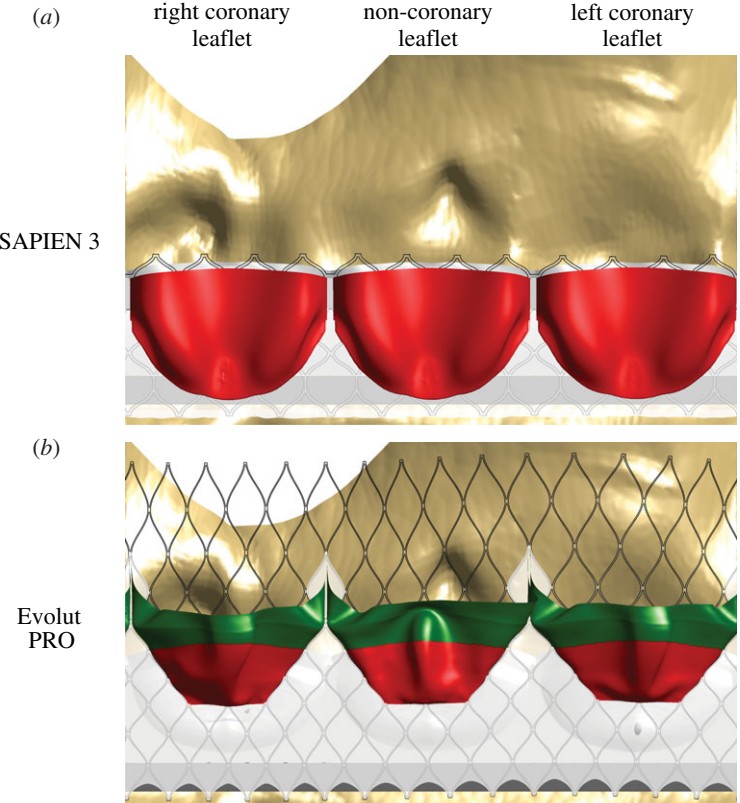

**Figure 3.** A spread view showing the leaflets' confinement in the SAPIEN (*a*) and Evolut (*b*) models. The confined regions of the TAVI leaflets are shown in red, while those that are not are displayed in green.

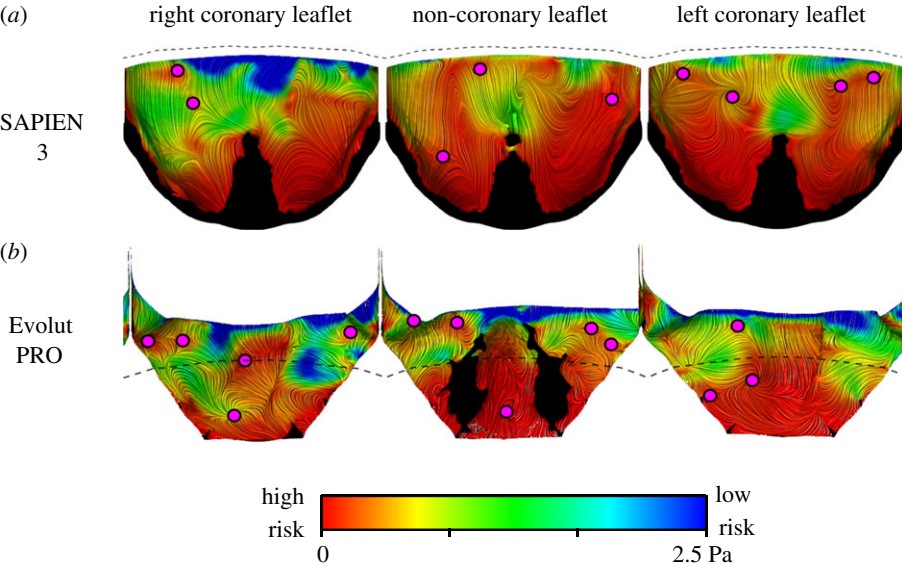

**Figure 4.** Comparison of the WSS streamlines, coloured by WSS magnitudes contours and WSSfp in the SAPIEN 3 (*a*) and Evolut PRO (*b*) models. Red colour indicates a high risk (low WSS magnitude), while blue colour points to low-risk locations and high shear stress. All the areas in contact with the valves' cuff and stent are shown in black and the dashed lines illustrate the leaflets' confinement by the surgical valve.

that in a global comparison, which ignores the spatial distribution of the results, the SAPIEN leaflets are exposed to a higher mean SC and ERT values than the Evolut valve (table 1). These unitless mean values, calculated as the weighted arithmetic mean to account for varying element sizes, are $2.48 \times 10^{-4}$ versus $11.0 \times 10^{-4}$ for the SC and $2.98 \times 10^{-4}$ versus $1.68 \times 10^{-4}$ for the ERT in the SAPIEN and Evolut models, respectively.

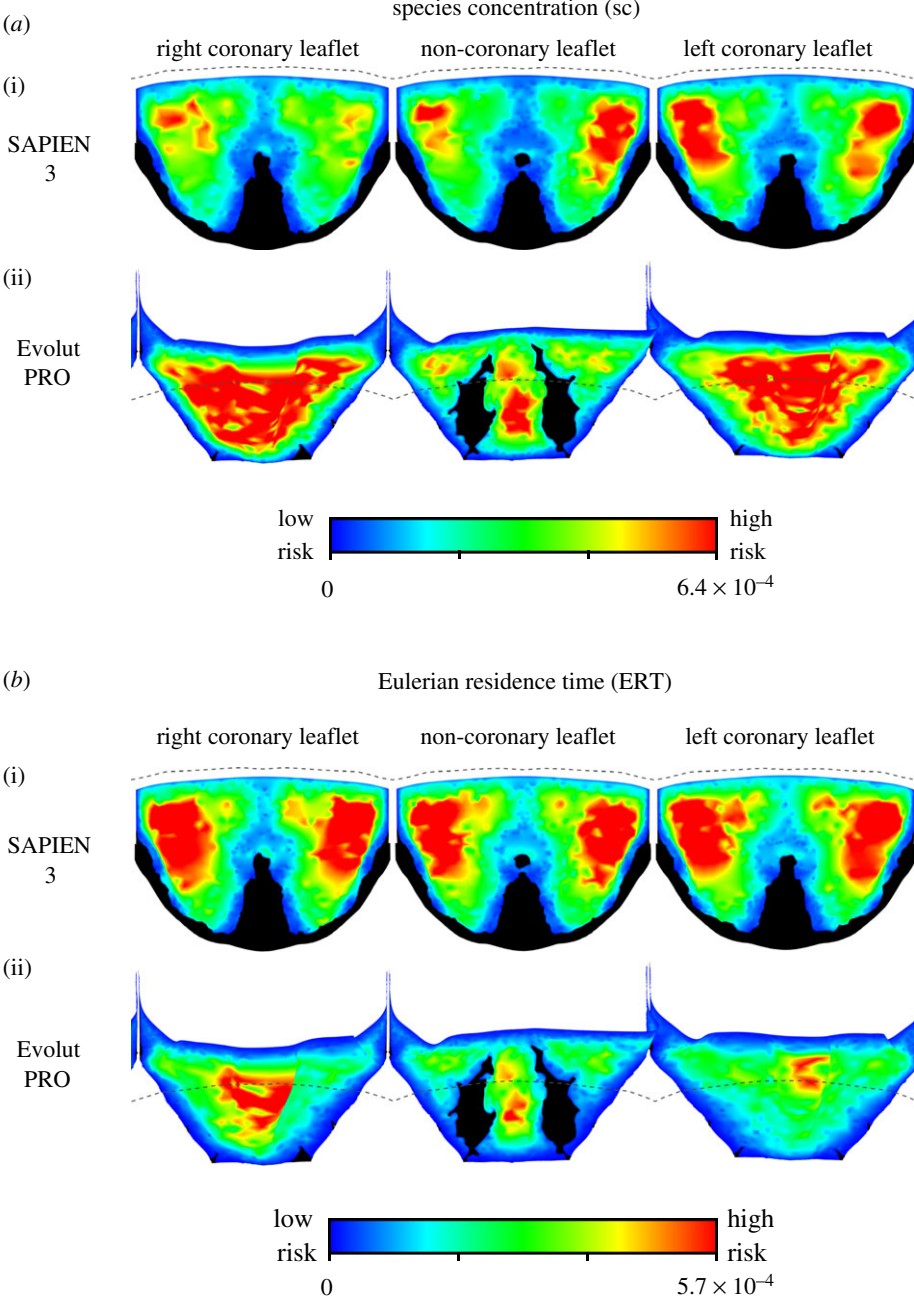

**Figure 5.** A comparison of the Eulerian thrombogenicity approaches for the two TAVI devices. The SC results for both SAPIEN and Evolut are shown in (*a*), while the results of the ERT approach are shown in (*b*). High-risk locations are shown in red while low-risk areas are shown in blue. The dashed lines illustrate the leaflets' confinement by the surgical valve.

### 3.1.4. Lagrangian approaches

The two Lagrangian approaches predicted similar risky areas for the two TAVI devices (figure 6). In both approaches and for the two device models, the stagnation regions are mainly concentrated at the proximal area of the leaflets. These areas seem to match the low WSS magnitude distribution (figure 4) better than the WSSfp or the Eulerian approaches (figures 4 and 5). In the Evolut model, the risky areas are mainly inside the fold of the non-coronary leaflet, while the risk of stagnation appears to be much lower in the right and left coronary leaflets. The SAPIEN leaflets cornered more particles than the Evolut (2.4% versus 1.25%) and lead to a higher number of vortices (9.27% versus 3.58%). Additionally, a supplementary video is provided to exemplify how the particle trajectories are able to indicate on stagnation regions (see the electronic supplementary material). The videos show green particles that are washed away with the pulsatile flow and red particles which get stuck in stagnation inside the neo-

**Table 1.** Quantitative comparison between the different thrombogenicity predictions for the post-ViV models of SAPIEN 3 and Evolut PRO devices.

| approach | WSSfp | | SC | | ERT | | PA | | VLI | |
|---|---|---|---|---|---|---|---|---|---|---|
| criterion | number of critical points | risky area (%) | mean | risky area (%) | mean | risky area (%) | number of cornered particles (%) | risky area (%) | number of vortices (%) | risky area (%) |
| SAPIEN 3 | 9 | 16.00 | $2.48 \times 10^{-4}$ | 5.11 | $2.98 \times 10^{-4}$ | 5.43 | 2.40 | 11.06 | 9.27 | 3.92 |
| Evolut PRO | 13 | 21.09 | $11.0 \times 10^{-4}$ | 10.57 | $1.68 \times 10^{-4}$ | 9.30 | 1.25 | 10.56 | 3.58 | 0.27 |

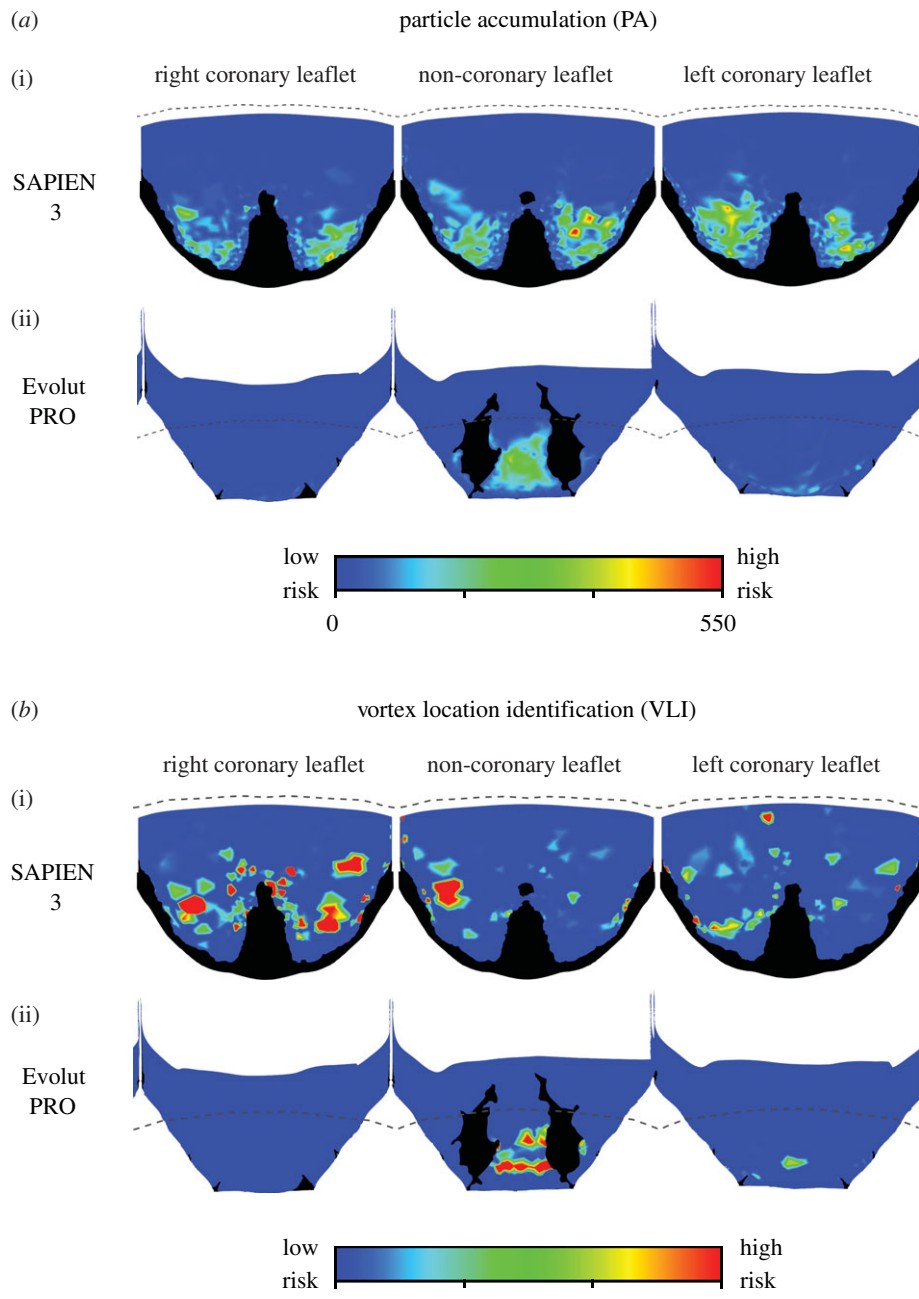

**Figure 6.** A comparison of the Lagrangian thrombogenicity approaches for the two TAVI devices. The PA model results for both SAPIEN and Evolut are shown in (*a*), while the results of the VLI approach are shown in (*b*). High-risk locations are shown in red while low-risk areas are shown in blue. The dashed lines illustrate the leaflets' confinement by the surgical valve.

sinuses. Note that the video was generated with a low frame rate from the solution, but with enough frame rate to visualize the stagnation region.

## 3.2. Quantitative comparison between the thrombogenic risk estimation models

A quantitative comparison of the thrombus formation risk was performed and is given in table 1. The number of WSSfp was higher for the Evolut TAVI than for the SAPIEN. Additionally, the high-risk area which was defined as the leaflets' area with the lowest 1% WSS magnitude was also higher in the Evolut than in the SAPIEN (21.09% versus 16%).

In the Eulerian approaches, the concept of a 'risky area' was defined as the sum of the entire surface nodes exposed to the top 1% magnitude of the SC or ERT. The nodes' 'areas' were chosen instead of that of the surface elements to account for the bias against non-homogeneous element size. Furthermore, the

**Figure 7.** Schematic representation of the risky areas, shown in red, predicted by the different thrombogenicity approaches for the two models. Note that the black areas are those in contact with other parts and the dashed lines illustrate the leaflets' confinement by the surgical valve.

risky area was normalized by the leaflet area. While the normalized risky area predicted by the SC and ERT approaches is similar, the SC results indicate a much higher mean concentration in the Evolut than in the SAPIEN, and the ERT indicates exactly the opposite trend with mean values in the SAPIEN almost doubling those of the Evolut (table 1).

In the Lagrangian approaches, the concept of a 'risky area' was defined as the sum of all the surface area elements with at least two particles (threshold used for PA) or a vortex (as defined by VLI) associated with them. This criterion predicted a small difference between the SAPIEN and Evolut in the PA approach, while the VLI predicted a risky area 10 times smaller in the Evolut than in the SAPIEN model. Still, both PA and VLI predicted a larger risky area in the SAPIEN model than in the Evolut, with a higher percentage of cornered particles near the leaflet or vortices.

Finally, the risky area predictions are based on the thresholds defined in table 1 and are shown in figure 7. There is a good agreement between the location predictions of the Eulerian approaches (SC and ERT). There is also a good agreement between the location predictions of the Lagrangian approaches (PA and VLI). While the predicted locations are different in these two groups, each one fits a different WSS parameter. The WSSfp is in agreement with the Eulerian approaches, and the WSS magnitude is with the Lagrangian approaches.

## 4. Discussion

In this study, five different approaches based on CFD simulations were employed to estimate the risk of a thrombus formation owing to flow stagnation after ViV implantation. These predictions were compared between two common TAVI devices: Evolut and SAPIEN. The aims of this work were to understand how these approaches' estimations differ and which TAVI device imposes a higher risk for leaflet thrombosis post-ViV implantations. Therefore, different criteria for the estimation of the thrombus high-risk locations were suggested and compared.

While all the five approaches assume that leaflet thrombosis is a result of stagnant flow in the neo-sinuses [12–17], the stagnation identification is based on different analyses of the flow field. The simplest approach analyses the WSS vectors on the surface of the leaflets, representing the features of vortical structures affecting the surface [24]. The WSS field can indicate thrombosis owing to either low WSS magnitude or the

location of singular points (WSSfp). However, each of these estimations predicted different high-risk locations. The locations of the WSSfp are in good agreement with the estimations of the Eulerian approaches, SC and ERT (figure 7), but differ from the reported thrombus locations of both *in vivo* and *in vitro* studies [12–17]. Possibly, these points indicate low advection therefore fitting the Eulerian approaches, which assumed an advection dominated transport [21]. On the other hand, the Lagrangian approaches (PA and VLI), which follow the particle trajectories, predicted higher risk locations at the proximal regions of the leaflets matching the low WSS magnitude regions in both TAVI models (figure 7). This correlation can be explained by the low velocities near the leaflets, resulting in both low shear and slow particle flow through the Lagrangian trajectories.

Out of the two predictions for thrombus location, the Lagrangian approaches and the low WSS estimations are in better agreement with the previously reported clinical [6] and *in vitro* experimental studies [12,13]. Additionally, these results better fit the expectation that the supra-annular Evolut should have lower thrombosis risk because of its partial confinement [17] (figure 3), which allows a better blood washout [32]. This advantage of the Evolut over the SAPIEN device is most noticeably seen when comparing the risky area predicted by the VLI approach (table 1). The fold in the non-coronary leaflet of the Evolut seems to be a significant cause for stagnation. Therefore, the results of the Lagrangian Evolut models seem to indicate lower thrombogenic risk considering that the current CFD models represent the worst-case scenario (systolic position). A more realistic case of unfixed leaflets is expected to reduce the thrombogenic risk even more.

Finally, the computational cost should be an additional consideration when choosing a thrombosis risk approach. The WSS analysis does not require additional post-processing calculations, making it the most cost-effective approach. The Eulerian approaches are more computationally expensive because they require the coupling and solving of additional PDEs. However, they do not match the expected high-risk locations. While the Lagrangian approaches can identify the stagnation locations, they are the most computationally expensive methods. Therefore, WSS magnitude analysis, which is also the most conservative approach (predicting the higher risky area of all approaches; table 1), was chosen as the preferred approach.

There are a few limitations in the current study that should be mentioned. This work did not model coronary flow, did not include fluid-structure interaction (FSI) modelling and was rather based only on CFD models, representing the worst-case scenario in terms of thrombogenic risk. However, there is an ongoing work to generate FSI models of ViV implantations to accurately predict the thrombogenic potential during a complete cardiac cycle. Furthermore, this work simulates the implantation of a 26 mm Evolut stent; however, the implantation of a 29 mm Evolut in a 27 mm Mitroflow stent has been simulated in an ongoing work, and because the resulting leaflets configuration is almost identical, similar predictions for leaflet thrombosis are expected in the bigger device. Additionally, the current study could not decisively recommend the most suitable approach for thrombus estimation. Further studies that test the capabilities of leaflet laceration (commonly known as BASILICA [40]) could aid in defining how the differences between the various approaches influence the estimation of leaflet thrombosis. Lastly, the lack of a suitable validation or insufficient data for comparison purposes imposes further constrains on our ability to recommend a specific approach. There are ongoing efforts to address these issues through *in vitro* study.

In summary, this work aimed to compare five numerical approaches for the estimation of leaflet thrombosis in two models of ViV implantations. The models included a Sorin Mitroflow surgical valve and either the Evolut or the SAPIEN TAVI device. Based on the predicted high-risk locations and the computation time, we concluded that the WSS magnitude is the most efficient approach to estimate the risk of leaflet thrombosis.

Data accessibility. Our dateset has been deposited to Dryad. For access, please follow the link: https://dx.doi.org/10.5061/dryad.31zcrjdjg [41].

Authors' contributions. R.P.M., A.F., S.C.S. and G.M contributed to the design of the study. R.P.M and H.Y developed all the models and analysed the data. All authors were involved in the preparation of the manuscript and agree on the submitted manuscript.

Competing interests. Prof. Finkelstein serves as a proctor, advisor and speaker for Medtronic Inc. and for Edwards Lifesciences. The remaining authors have no conflict of interest to declare.

Funding. This work was partially supported by grants from the Raymond and Beverly Sackler Fund for Convergence Research in Biomedical, Physical and Engineering Sciences and from the Nicholas and Elizabeth Slezak Super Centre for Cardiac Research and Biomedical Engineering at Tel Aviv University.

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
