## [Reviewer comments · Royal Society Open Science]

Review History

RSOS-201838.R0 (Original submission)

Review form: Reviewer 1

Is the manuscript scientifically sound in its present form?

Yes

Are the interpretations and conclusions justified by the results?

Yes

Is the language acceptable?

Yes

Do you have any ethical concerns with this paper?

No

Have you any concerns about statistical analyses in this paper?

No

Recommendation?

Accept as is

Comments to the Author(s)

All of my original concerns have been addressed by the authors.

Review form: Reviewer 2

Is the manuscript scientifically sound in its present form?

Yes

Are the interpretations and conclusions justified by the results?

Yes

Is the language acceptable?

Yes

Do you have any ethical concerns with this paper?

No

Have you any concerns about statistical analyses in this paper?

No

Recommendation?

Accept with minor revision (please list in comments)

Comments to the Author(s)

The authors proposed a revised version of their numerical work answering to the doubt and the criticisms made by the reviewer. Nevertheless, one aspect is still unclear.

If the reviewer has understood correctly for both the devices (i.e the Evolut and the Sapien valve) the leaflets open geometry is achieved through the same strategy, that is the application of a uniform pressure ramp onto the ventricular leaflet surface. The reviewer finds reasonable the geometry obtained for the Sapien valve, since the device geometry is symmetric. Nevertheless, some perplexities arise from the shape assumed by the leaflets of the Evolut device which is markedly not symmetric. Is this problem related to the deformed geometry of the stent or to the mechanical properties of the leaflets? Please add some comments also regarding the constitutive behavior prescribed in the model.

Decision letter (RSOS-201838.R0)

Dear Dr Plitman Mayo

On behalf of the Editors, we are pleased to inform you that your Manuscript RSOS-201838 "Numerical Models for Assessing the Risk of Leaflet Thrombosis Post-Transcatheter Aortic Valve-in-Valve Implantation" has been accepted for publication in Royal Society Open Science subject to

minor revision in accordance with the referees' reports. Please find the referees' comments along with any feedback from the Editors below my signature.

Please submit your revised manuscript and required files (see below) no later than 7 days from today's (ie 16-Nov-2020) date. Note: the ScholarOne system will 'lock' if submission of the revision is attempted 7 or more days after the deadline. If you do not think you will be able to meet this deadline please contact the editorial office immediately.

on behalf of Prof R. Kerry Rowe (Subject Editor)
openscience@royalsociety.org

Associate Editor Comments to Author:

Well done on thoroughly addressing the referees' concerns in this transferred paper - only one comment remains from one of the reviewers, and we would like you to tackle this before a final acceptance can be issued. Thanks in advance.

Reviewer comments to Author:

Reviewer: 1

Comments to the Author(s)

All of my original concerns have been addressed by the authors.

Reviewer: 2

Comments to the Author(s)

The authors proposed a revised version of their numerical work answering to the doubt and the criticisms made by the reviewer. Nevertheless, one aspect is still unclear.

If the reviewer has understood correctly for both the devices (i.e the Evolut and the Sapien valve) the leaflets open geometry is achieved through the same strategy, that is the application of a uniform pressure ramp onto the ventricular leaflet surface. The reviewer finds reasonable the geometry obtained for the Sapien valve, since the device geometry is symmetric. Nevertheless, some perplexities arise from the shape assumed by the leaflets of the Evolut device which is markedly not symmetric. Is this problem related to the deformed geometry of the stent or to the

mechanical properties of the leaflets? Please add some comments also regarding the constitutive behavior prescribed in the model.

===PREPARING YOUR MANUSCRIPT===

===PREPARING YOUR REVISION IN SCHOLARONE===

- An individual file of each figure (EPS or print-quality PDF preferred [either format should be produced directly from original creation package], or original software format).
 - An editable file of each table (.doc, .docx, .xls, .xlsx, or .csv).
 - An editable file of all figure and table captions.
- Note: you may upload the figure, table, and caption files in a single Zip folder.
- Any electronic supplementary material (ESM).
 - If you are requesting a discretionary waiver for the article processing charge, the waiver form must be included at this step.
 - If you are providing image files for potential cover images, please upload these at this step, and inform the editorial office you have done so. You must hold the copyright to any image provided.
 - A copy of your point-by-point response to referees and Editors. This will expedite the preparation of your proof.

- Ensure that your data access statement meets the requirements at <https://royalsociety.org/journals/authors/author-guidelines/#data>. You should ensure that you cite the dataset in your reference list. If you have deposited data etc in the Dryad repository, please only include the 'For publication' link at this stage. You should remove the 'For review' link.
- If you are requesting an article processing charge waiver, you must select the relevant waiver option (if requesting a discretionary waiver, the form should have been uploaded at Step 3 'File upload' above).
- If you have uploaded ESM files, please ensure you follow the guidance at <https://royalsociety.org/journals/authors/author-guidelines/#supplementary-material> to include a suitable title and informative caption. An example of appropriate titling and captioning may be found at https://figshare.com/articles/Table_S2_from_Is_there_a_trade-off_between_peak_performance_and_performance_breadth_across_temperatures_for_aerobic_scope_in_teleost_fishes_/3843624.

Author's Response to Decision Letter for (RSOS-201838.R0)

See Appendix A.

Decision letter (RSOS-201838.R1)

Dear Dr Plitman Mayo,

It is a pleasure to accept your manuscript entitled "Numerical Models for Assessing the Risk of Leaflet Thrombosis Post-Transcatheter Aortic Valve-in-Valve Implantation" in its current form for publication in Royal Society Open Science.

on behalf of Professor R. Kerry Rowe (Subject Editor)
openscience@royalsociety.org

Appendix A

Response to Decision Letter – Manuscript ID RSOS – 201838

Dear Prof. Rowe,

We thank you for accepting our manuscript in the Journal of the Royal Society Open Science. We have addressed the remained comment of Reviewer 2, our answer is given below and the changes are highlighted in the final submission draft. We also changed the reference style to Vancouver with DOI for all the journal articles.

Answer to Reviewer:

If the reviewer has understood correctly for both the devices (i.e the Evolut and the Sapien valve) the leaflets open geometry is achieved through the same strategy, that is the application of a uniform pressure ramp onto the ventricular leaflet surface. The reviewer finds reasonable the geometry obtained for the Sapien valve, since the device geometry is symmetric. Nevertheless, some perplexities arise from the shape assumed by the leaflets of the Evolut device which is markedly not symmetric. Is this problem related to the deformed geometry of the stent or to the mechanical properties of the leaflets? Please add some comments also regarding the constitutive behavior prescribed in the model.

We thank the reviewer for the comment. The deployed configuration of the Evolut is greatly influenced by the arch of the ascending aorta, leading to an asymmetric deployed configuration. As a result, the open configuration of the Evolut leaflets is also asymmetric. We have added a sentence clarifying this issue.

Additionally, we have added to the manuscript that the material model of the leaflets was defined as linear elastic.